# PROVABLE DISTRIBUTIONAL VALUE ITERATION UNDER PARTIAL OBSERVABILITY

## ABSTRACT

In many real-world planning tasks, agents must tackle uncertainty about the environment's state and variability in the outcomes induced by stochastic dynamics and rewards. Motivated by recent progress in world model approaches—where latent models approximate beliefs and support planning—we extend Distributional Reinforcement Learning (DistRL), which models the entire return distribution for fully observable domains, to Partially Observable Markov Decision Processes (POMDPs). Concretely, we introduce new distributional Bellman operators for partial observability and prove their convergence under the supremum $p$-Wasserstein metric. We also propose a finite representation of these return distributions via $\psi$-vectors, generalizing the classical $\alpha$-vectors in POMDP solvers. Building on this, we develop Distributional Point-Based Value Iteration (DPBVI), which integrates $\psi$-vectors into a standard point-based backup procedure—*bridging DistRL and POMDP planning*. Our experiments demonstrate that DPBVI recovers classical Point-Based Value Iteration (PBVI) in the risk-neutral case, validating the distributional extension.

## 1 INTRODUCTION

Reinforcement Learning (RL) traditionally maximizes expected returns, treating each future reward distribution as a scalar. However, growing evidence suggests that modeling the full return distribution, rather than just its mean, can yield robust policies, better exploration, and richer theoretical insights (Bellemare et al., 2023). This distributional perspective, referred to as Distributional Reinforcement Learning (DistRL), captures variability and higher-order statistics of returns.

While DistRL has been well-studied in fully observable Markov Decision Processes (MDPs), many real-world systems are partially observable (Chiam et al., 2024; Komorowski et al., 2018; Caballero-Martin et al., 2024), forcing the agent to infer the latent state from noisy or incomplete observations. Such problems are naturally framed as Partially Observable Markov Decision Processes (POMDPs), but the literature on distributional methods in these settings remains sparse.

Recently, deep model-based planning methods—often referred to as world models—have shown impressive success (Hafner et al., 2025; Zhang et al., 2023). These models jointly learn dynamics and a latent belief state. We are further motivated by the success of deep DistRL methods such as C51, IQN, QR-DQN (Bellemare et al., 2017; Dabney et al., 2018a;b).

In this work, we present a formal extension of DistRL to POMDPs, bridging the gap between distributional theory and partial observability. Concretely, we make three major contributions:

- **Formal Extension of DistRL to POMDPs.** We define the partially observable distributional evaluation operator $\widetilde{\mathcal{T}}_{PO}$ and its optimality operator $\widetilde{\mathcal{T}}_{PO}^*$, showing that both are $\gamma$-contractions under the supremum $p$-Wasserstein metric ($1 \le p < \infty$). This result generalizes classical DistRL convergence results to the partially observable setting.

- **Finite Representation via $\psi$-Vectors.** We introduce $\psi$-vectors, a distributional analog to the well-known $\alpha$-vectors in POMDP theory. In the risk-neutral regime, we show that a finite set of $\psi$-vectors suffices to represent the optimal distributional value function while preserving the piecewise linear and convex (PWLC) property in the Wasserstein space.

- **Distributional Point-Based Value Iteration (DPBVI).** We adapt Point-Based Value Iteration (PBVI) to the distributional setting, yielding DPBVI, which uses $\psi$-vectors instead of $\alpha$-vectors. Under risk-neutral objectives, DPBVI converges to the same policy as PBVI yet lays the groundwork for risk-sensitive extensions by maintaining full return distributions.

Alongside these theoretical results, we release our source code[1] to facilitate further exploration of distributional approaches in partially observable domains. To our knowledge, this is the first work to establish DistRL theory in partially observable domains.

## 2 RELATED WORK

**POMDP Planning** The extension of MDPs to partially observable settings was introduced by Åström (1965), who showed that beliefs are sufficient planning statistics and that any POMDP can be reformulated as a belief MDP. Sondik's thesis Sondik (1971) later established that optimal control is achievable in both finite and discounted infinite-horizon settings, introducing $\alpha$-vectors and proving that the optimal value function is piecewise linear and convex (PWLC), and thus representable by a finite set of $\alpha$-vectors. Despite this structure, POMDP planning remains intractable: planning for the entire belief space is PSPACE-complete in the finite-horizon case and undecidable in the discounted infinite-horizon case (Hsu et al., 2007). This has motivated approximate solvers that restrict planning to reachable beliefs, such as Point-Based Value Iteration (PBVI) (Pineau et al., 2003). Our work builds on this line by extending point-based methods to the distributional setting, where planning is guided by return distributions rather than scalar expectations.

**DistRL** Bellemare et al. introduced DistRL in Bellemare et al. (2017), establishing contraction results and demonstrating state-of-the-art performance on Atari 2600 benchmarks. Follow-up work analyzed categorical (Rowland et al., 2018) and quantile (Dabney et al., 2018b) approximations, and later proposed implicit quantile networks (IQN) to approximate the continuous quantile function (Dabney et al., 2018a). This line of research has since been consolidated into a unified theoretical framework (Bellemare et al., 2023). While prior work has focused on fully observable MDPs, little attention has been given to partially observable settings. In this paper, we establish contraction and finite-representation results for distributional planning in POMDPs. This represents the first extension of DistRL beyond fully observable MDPs to the POMDP setting.

## 3 SETTING

We consider a finite, discrete-time Partially Observable Markov Decision Process (POMDP) $\mathcal{P}$. POMDPs are defined as a tuple: $\langle \mathcal{S}, \mathcal{A}, \mathcal{O}, T, \Omega, b_0, P_R, \gamma \rangle$, where $\mathcal{S}$ is a set of discrete states, $\mathcal{A}$ is a set of discrete actions, $\mathcal{O}$ is the discrete space of noisy and/or incomplete state information, $T(s, a, s') = P(s_{t+1} = s' \mid s_t = s, a_t = a)$ is the state transition model, $\Omega(o', s', a) = P(o_{t+1} = o' \mid s_{t+1} = s', a_t = a)$ is the sensor model, $b_0 \in \Delta$ is the initial belief, $P_R(\cdot \mid s, a)$ is the conditional distribution of the immediate reward $r$ when taking action $a$ in state $s$, and $0 \leq \gamma < 1$ is the discount factor. We denote the expected immediate reward by $\mathcal{R}(s, a) = \mathbb{E}_{R \sim P_R(\cdot \mid s, a)}[R]$.

In this setting, the agent is unable to directly observe the state of the environment and must rely on a developed belief state $b \in \Delta$, a probability distribution across $\mathcal{S}$. The belief state serves as sufficient information for the agent to behave optimally (Åström, 1965; Kaelbling et al., 1998). The agent's belief is developed using the sequence of observations $b_t = P(s_t \mid b_0, a_0, o_1, \ldots, o_{t-1}, a_{t-1}, o_t)$ by

$$
\begin{aligned}
b_t(s_t) &= \tau(b_{t-1}, a, o) \\
&= \frac{\Omega(o_t, s_t, a_{t-1}) \sum_{s_{t-1} \in \mathcal{S}} T(s_{t-1}, a_{t-1}, s_t) b_{t-1}(s_{t-1})}{\sum_{s_t \in \mathcal{S}} \Omega(o_t, s_t, a_{t-1}) \sum_{s_{t-1} \in \mathcal{S}} T(s_{t-1}, a_{t-1}, s_t) b_{t-1}(s_{t-1})}
\end{aligned}
\tag{1}
$$

In POMDPs, the objective is to learn an optimal, stationary policy $\pi^*(a \mid b)$ which maximizes the expected return $\mathbb{E}\left[\sum_{t=0}^{T} \gamma^t \mathcal{R}(s_t, a_t)\right]$ for time horizon $T \in \mathbb{N}^+ \cup \{\infty\}$. Policies in this setting may be viewed as conditional plans and are typically learned by learning the optimal value function

---

[1]Source code available at *redacted for double-blind review*.

$$V^*(b) = \max_{a \in \mathcal{A}} \left[ \sum_{s \in \mathcal{S}} \mathcal{R}(s, a)b(s) + \gamma \sum_{o' \in \mathcal{O}} \sum_{s' \in \mathcal{S}} \Omega(o', s', a) \sum_{s \in \mathcal{S}} T(s, a, s')b(s)V^*(b') \right] \quad (2)$$

The $n$-th horizon value function is comprised of a set of $\alpha$-vectors $V_n = \{\alpha_0, \alpha_1, \dots, \alpha_m\}$ and is piecewise linear and convex (PWLC) (Sondik, 1978). Each $\alpha$-vector is a $|\mathcal{S}|$-dimensional hyperplane and defines the value function for some bounded region of $\Delta$ (i.e., $\max_{\alpha \in V} \alpha \cdot b$). At each planning step, the next value function $V_n$ may be computed from the previous value function $V_{n-1}$ via the Bellman optimality operator $\mathcal{T}_{PO}$

$$V_n(b) = \mathcal{T}_{PO} V_{n-1}$$
$$= \max_{a \in \mathcal{A}} \left[ \sum_{s \in \mathcal{S}} \mathcal{R}(s, a)b(s) + \gamma \sum_{o' \in \mathcal{O}} \max_{\alpha \in V_{n-1}} \sum_{s' \in \mathcal{S}} \Omega(o', s', a) \sum_{s \in \mathcal{S}} T(s, a, s')b(s)\alpha(s') \right] \quad (3)$$

### 3.1 POINT-BASED VALUE ITERATION

PBVI computes $V_t = \mathcal{T}_{PO} V_{t-1}$ in three steps. First, it creates projections for each action and observation

$$\Gamma^{a,*} \leftarrow \alpha^{a,*}(s) = \mathcal{R}(s, a)$$
$$\Gamma^{a,o'} \leftarrow \alpha_i^{a,o'}(s) = \gamma \sum_{s' \in \mathcal{S}} T(s, a, s')\Omega(o', s', a)\alpha_i^{t-1}(s'), \forall \alpha_i^{t-1} \in V_{t-1} \quad (4)$$

Next, the value of action $a$ at belief $b$ is computed by

$$\Gamma_b^a = \Gamma^{a,*} + \sum_{o' \in \mathcal{O}} \arg\max_{\alpha \in \Gamma^{a,o'}} (\alpha \cdot b) \quad (5)$$

Lastly, the best action at each belief is used to construct $V_t$

$$V_t \leftarrow \arg\max_{\Gamma_b^a, \forall a \in \mathcal{A}} (\Gamma_b^a \cdot b), \forall b \in \mathcal{B} \quad (6)$$

$V_t$ is comprised of at most $|\mathcal{B}|$ $\alpha$-vectors, thus requiring $|\mathcal{S}||\mathcal{A}||V_{t-1}||\mathcal{O}||\mathcal{B}|$ operations per backup.

### 3.2 DISTRIBUTIONAL REINFORCEMENT LEARNING

Traditionally, the value function models the return expectation. DistRL, instead, aims to learn the compound distribution of the returns $Z^\pi(s)$ sourced to randomness in (1) reward $R$ (2) transition $P^\pi$, and (3) distribution of the next-state value $Z(S')$. Let $\widetilde{\mathcal{T}}$ be the distributional Bellman operator such that

$$\widetilde{\mathcal{T}}^\pi Z(s) \stackrel{D}{=} R + \gamma P^\pi Z(s) \quad (7)$$

where $\stackrel{D}{=}$ denotes equality in distribution, $R$ is the reward random variable, and $P^\pi Z(s) \stackrel{D}{=} Z(S')$ for $S' \sim T(s, a, \cdot)$.

The distributional Bellman operator has been shown to be a $\gamma$-contraction under the supremum $p$-Wasserstein metric space $\forall p \in [1, \infty]$ (Bellemare et al., 2023). The distributional Bellman optimality operator $\widetilde{\mathcal{T}}^*$, which selects actions that maximize the expected return, similarly converges to the fixed point under the supremum $p$-Wasserstein metric space $\forall p \in [1, \infty]$ (Bellemare et al., 2023) if there is a unique optimal policy and a mean-preserving distribution representation $\mathcal{F}$ is used. In the case of the existence of multiple optimal policies, however, convergence isn't guaranteed because differing distributions can be similar in expectation.

# 4 DISTRIBUTIONAL REINFORCEMENT LEARNING UNDER PARTIAL OBSERVABILITY

In the presence of partial observability we must account for additional sources of randomness; namely, randomness in (a) the unobservable state $\mathcal{S}$, (b) sensing $\Omega$. Therefore, the return distribution is learned with respect to belief instead of state (i.e., $Z(b)$).

## 4.1 THE PARTIALLY OBSERVABLE DISTRIBUTIONAL BELLMAN OPERATORS

Let $\widetilde{\mathcal{T}}_{PO}$ be the partially observable distributional Bellman operator such that

$$\widetilde{\mathcal{T}}_{PO}^{\pi} Z(b) \stackrel{D}{=} R + \gamma \tau^{\pi} Z(b) \tag{8}$$

where $\tau^{\pi} Z(b) \stackrel{D}{=} Z(B')$ for $S \sim b$, $A \sim \pi(\cdot \mid b)$, $S' \sim T(\cdot \mid S, A)$, $O \sim \Omega(\cdot \mid S', A)$, $B' \sim \tau(b, A, O)$. The partially observable distributional Bellman optimality operator $\widetilde{\mathcal{T}}_{PO}^{*}$ may be similarly defined, but where actions are selected to maximize the expected return. The distributional Bellman operators in the partially observable setting share convergence properties with those of the fully observable setting, because any POMDP can be rewritten as a belief MDP, where the state of the belief MDP is the belief of the POMDP (Kaelbling et al., 1998).

We denote by $\mathfrak{P}_p(\mathbb{R})$ the set of Borel probability measures on $\mathbb{R}$ with finite $p$-th moment, ensuring that the $p$-Wasserstein distance is well-defined. The space $\mathfrak{P}_p(\mathbb{R})^{\Delta}$ then denotes the set of functions $\eta : \Delta \to \mathfrak{P}_p(\mathbb{R})$, where $\Delta$ is the belief space of the POMDP.

**Theorem 1.** *Let $p \in [1, \infty)$ and $\gamma \in [0, 1)$. Consider a POMDP $\mathcal{P}$ in which each reward distribution $P_R(\cdot \mid s, a)$ belongs to $\mathfrak{P}_p(\mathbb{R})$ for all $(s, a) \in \mathcal{S} \times \mathcal{A}$. Then the partially observable distributional Bellman operator $\widetilde{\mathcal{T}}_{PO}^{\pi}$ is a $\gamma$-contraction on $\mathfrak{P}_p(\mathbb{R})^{\Delta}$ in the supremum $p$-Wasserstein distance $\bar{w}_p$. That is,*

$$\bar{w}_p(\widetilde{\mathcal{T}}_{PO}^{\pi} \eta, \widetilde{\mathcal{T}}_{PO}^{\pi} \eta') \leq \gamma \bar{w}_p(\eta, \eta'),$$

*$\forall \eta, \eta' \in \mathfrak{P}_p(\mathbb{R})^{\Delta}$.*

Theorem 1 follows from the standard $\gamma$-contraction properties of $\widetilde{\mathcal{T}}^{\pi}$. Because $\widetilde{\mathcal{T}}_{PO}^{\pi}$ is a contraction mapping, by the Banach Fixed-Point Theorem, there exists a unique $\eta^* \in \mathfrak{P}_p(\mathbb{R})^{\Delta}$ such that $\eta^*$ is a fixed point. *See Appendix A.1 for proof.*

**Theorem 2.** *Let $p \in [1, \infty)$ and let $\widetilde{\mathcal{T}}_{PO}^{*}$ be the partially observable distributional Bellman optimality operator for a POMDP $\mathcal{P}$. Assume that for every $(s, a) \in \mathcal{S} \times \mathcal{A}$, the reward distribution $P_R(\cdot \mid s, a)$ belongs to $\mathfrak{P}_p(\mathbb{R})$. Suppose there is a unique optimal policy $\pi^*$. Then, for any initial return-distribution function $\eta_0 \in \mathfrak{P}_p(\mathbb{R})^{\Delta}$, the sequence of iterates*

$$\eta_{k+1} = \widetilde{\mathcal{T}}_{PO}^{*} \eta_k$$

*converges in the supremum $p$-Wasserstein distance $\overline{w}_p$ to $\eta^{\pi^*}$, the return distribution associated with the unique optimal policy.*

**Corollary 1.** *Consider a mean-preserving projection $\Pi_{\mathcal{F}}$ for some representation $\mathcal{F}$. Suppose $\Pi_{\mathcal{F}}$ is a nonexpansion in the supremum $p$-Wasserstein distance, and let the finite domain $\mathfrak{P}_p(\mathbb{R})^{\Delta}$ be closed under $\Pi_{\mathcal{F}}$. If there is a unique optimal policy $\pi^*$ then, under the conditions of Theorem 2, the sequence*

$$\eta_{k+1} = \Pi_{\mathcal{F}} \widetilde{\mathcal{T}}_{PO}^{*} \eta_k$$

*converges to the fixed point $\eta^{\pi^*}$.*

These results build on the convergence guarantees of the distributional Bellman optimality operator in (Bellemare et al., 2023, Theorem 7.9), which assumes a unique optimal policy and analyzes the tabular MDP setting. Our proof adapts these arguments to the belief MDP by ensuring the required continuity and compactness conditions hold, allowing the result to generalize to partially observable settings (see Appendix A.2). Bellemare et al. note that the unique optimal policy assumption is necessary for contraction guarantees but they empirically observe that algorithms like C51 often perform well even when this condition may not strictly hold Bellemare et al. (2023).

Theorem 2 states if there is a unique optimal policy, repeatedly applying $\widetilde{\mathcal{T}}_{PO}^*$ will converge to the optimal return distribution function $\eta$. Corollary 1 establishes that if we approximate the return distributions after each greedy update, the resulting approximate iteration still converges to the same fixed point, provided there is a unique optimal policy.

## 4.2 $\psi$-VECTORS

In classical POMDP value iteration, each conditional plan—a sequence of actions contingent on future observations—is captured by an $\alpha$-vector, which stores said plan's expected return for each state. Thus, a belief-state's value is found by selecting the $\alpha$-vector with the highest dot product $\alpha \cdot b$. While this approach gives the expected returns of each plan, it does not capture the uncertainty around possible outcomes.

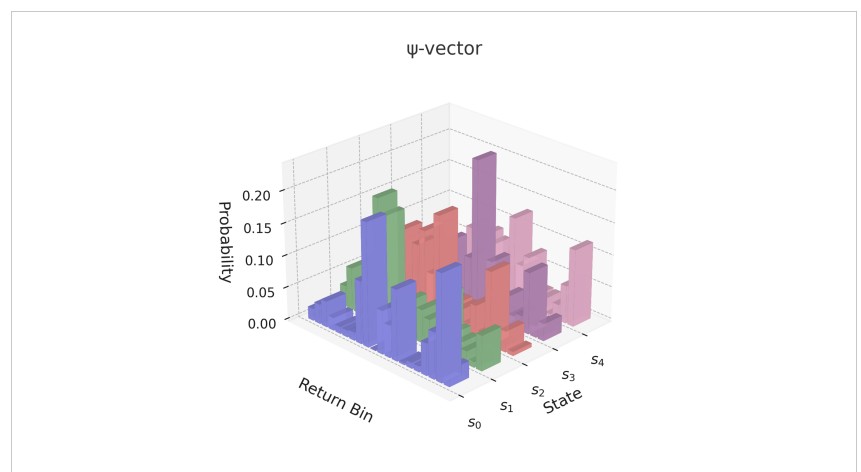

Figure 1: A $\psi$-vector stores a return distribution for each state, generalizing classical $\alpha$-vectors. The figure shows the *categorical* approximation $\hat{\psi}$ used in DPBVI (Section 5); the theoretical framework (Section 4.2) uses true return distributions.

To address this, we adopt a distributional viewpoint: rather than storing a scalar expectation, we store a distribution over returns for each plan. We call these $\psi$-vectors (Figure 1). Formally, a $\psi$-vector is

$$\Psi = (\psi^s)_{s \in \mathcal{S}}, \tag{9}$$

where $\psi^s \in \mathfrak{P}_p(\mathbb{R})$ is a distribution over returns conditioned on being in state $s$. Given a belief $b \in \Delta$, we define the (risk-neutral) expected return of $\Psi$ at $b$ as

$$\langle \Psi, b \rangle = \sum_{s \in \mathcal{S}} b(s) \mathbb{E}[\psi^s] \tag{10}$$

Given a set of candidate $\psi$-vectors $\Gamma$, the distributional value function at a belief $b$ is then obtained by first selecting the $\psi$-vector with maximal expected return:

$$\Psi^*(b) = \arg\max_{\Psi \in \Gamma} \langle \Psi, b \rangle \tag{11}$$

The corresponding return distribution at belief $b$ is then defined as the belief-weighted mixture of the state-conditioned distributions:

$$Z(b) = \sum_{s \in \mathcal{S}} b(s) \psi^{*,s}, \tag{12}$$

where $\psi^{*,s}$ is the $s$th component of $\Psi^*(b)$.

**Theorem 3.** *Under the assumptions of Theorem 2, the optimal distributional value function $Z^*(b)$ admits a finite piecewise-linear and convex (PWLC) representation under belief-linear mixture with*

*respect to supremum p-Wasserstein $\bar{w}_p$. Specifically, there exists a finite set of $\psi$-vectors $\Gamma$ such that for any belief state $b \in \Delta$,*

$$Z^*(b) = \sum_{s \in \mathcal{S}} b(s)\psi^{*,s},$$

*where $\Psi^*(b) \in \Gamma$ is the maximizing $\psi$-vector at belief $b$ and $\psi^{*,s}$ denotes its sth component.*

*See Appendix A.3 for proof.*

**Corollary 2.** *Under the assumptions of Theorem 2 and Corollary 1, let $\hat{\Psi}$ denote the image of $\Psi$ under a mean-preserving $\bar{w}_p$-nonexpansive projection $\Pi_{\mathcal{F}}$ applied componentwise to each $\psi^s$. Then the projected optimal distributional value function $\hat{Z}^*(b)$ admits a finite PWLC representation in the supremum p-Wasserstein metric, using a finite set of projected $\hat{\psi}$-vectors.*

By Theorem 3, $Z^*$ inherits the PWLC structure of classical POMDPs, but within the Wasserstein metric space. A finite set of $\psi$-vectors suffices to represent $Z^*$, directly mirroring the finite $\alpha$-vector representation of $V^*$.

Corollary 2 further ensures that any mean-preserving, $\bar{w}_p$-nonexpansive projection preserves this finiteness. That is, the projected optimal value function $\hat{Z}^*$ is still representable by a finite set of projected $\hat{\psi}$-vectors. Thus, tractability is maintained even under practical finite distributional approximations.

Our proofs focus on the discounted infinite-horizon case, where contraction guarantees convergence to a unique fixed point. Since POMDPs admit finite $\alpha$-vector representations in both finite and discounted infinite-horizon settings (Sondik, 1971), our results apply to both.

## 5 DISTRIBUTIONAL POINT-BASED VALUE ITERATION

Building upon PBVI and DistRL in the partially observable setting, we now introduce *Distributional Point-Based Value (DPBVI)*. Conceptually, DPBVI is PBVI with $\widetilde{\mathcal{T}}_{PO}^*$ in place of $\mathcal{T}_{PO}^*$. As in standard PBVI, we maintain a finite set of belief points $\mathcal{B} \subset \Delta$, perform point-based backups (Pineau et al., 2003), and update one vector per belief. The key difference is that DPBVI maintains a finite collection of $\hat{\psi}$-vectors rather than $\alpha$-vectors.

In PBVI, each $\alpha$-vector encodes expected returns for a conditional plan. In DPBVI, each $\hat{\psi}$-vector stores a return distribution for each state, capturing the full distributional return of a conditional plan.

By Corollary 2, any mean-preserving and $\bar{w}_p$-nonexpansive projection admits a finite PWLC representation via projected $\hat{\psi}$-vectors. In practice, we use a categorical parameterization to instantiate $\hat{\psi}$-vectors for computational tractability. This approximation is mean-preserving but is not assumed to satisfy the nonexpansion conditions of Corollaries 1–2; it is adopted purely as a practical representation of return distributions.

Unlike PBVI, DPBVI assumes stochastic rewards. In practice, we approximate the reward distribution $P_R(\cdot \mid s, a)$ using a categorical representation, denoted the approximation by $\hat{R}(\cdot \mid s, a)$. Note the support of $\hat{R}$ need not match the support used for the $\hat{\psi}$-vector representation.

### 5.1 THE DPBVI BACKUP

Given the previous set of $\hat{\psi}$-vectors $\Gamma^{t-1}$, each DPBVI backup follows four steps: 1. action-observation projection, 2. categorical distribution projection, 3. belief-specific maximization, and 4. candidate $\hat{\psi}$-vector selection.

For each action $a \in \mathcal{A}$, observation $o' \in \mathcal{O}$, and $\hat{\Psi} \in \Gamma^{t-1}$, we construct a candidate (un-normalized) $\hat{\psi}$-vector $\hat{\Psi}_{a,o'}^t$ with components $\hat{\psi}_{a,o'}^{t,s}$ for each state $s \in \mathcal{S}$ via

$$\hat{\psi}_{a,o'}^{t,s} = \sum_{s' \in \mathcal{S}} T(s, a, s') \Omega(o', s', a) \left( \hat{R}(\cdot \mid s, a) \oplus_\gamma \hat{\psi}^{s'} \right), \tag{13}$$

where $\mu \oplus_\gamma \nu$ denotes the distribution of $R + \gamma Z$. In practice, this operation is implemented by:

- iterating over reward atoms $r$ with probability $\hat{R}(r \mid s, a)$ and return atoms $z$ with probability $\hat{\psi}^{s'}(z)$,

- and adding probability mass $T(s, a, s')\Omega(o', s', a)\hat{R}(r \mid s, a)\hat{\psi}^{s'}(z)$ to the categorical bin corresponding to $r + \gamma z$, after projecting $r + \gamma z$ onto the fixed support using the categorical projection operator $\Pi_c$.

Collecting all such candidates over $\hat{\Psi} \in \Gamma^{t-1}$ yields the set $\Gamma_a^{o',t}$ of action-observation $\hat{\psi}$-vectors. This step yields $|\mathcal{A}| \times |\mathcal{O}| \times |\Gamma^{t-1}|$ projected distributions, denoted by $\Gamma^{,t}$. Because each distribution reflects one action-observation pair, it may be *subnormalized* until we later sum over observations.

For each belief $b \in \mathcal{B}$, we select, for every action-observation pair, the $\hat{\psi}$-vector with the highest expected return under $b$. For a fixed action $a$ and observation $o'$, define

$$\hat{\Psi}_{a,o'}^{b,*} = \arg\max_{\hat{\Psi} \in \Gamma_a^{o',t}} \langle \hat{\Psi}, b \rangle. \tag{14}$$

We then aggregate these best per-observation candidates into an action-specific $\hat{\psi}$-vector:

$$\hat{\Psi}_a^b = \sum_{o' \in \mathcal{O}} \hat{\Psi}_{a,o'}^{b,*}. \tag{15}$$

This mirrors the cross-sum and maximize operation in PBVI: for each action, we keep the best projected $\hat{\psi}$-vector per observation and sum across observations to recover a full next-step return distribution for that action at belief $b$. We store this set of maximizing $\hat{\psi}$-vectors in $\Gamma_a^{b,t}$.

Finally, for each belief $b$, we pick the best action's distribution by

$$\Psi_b^t \leftarrow \arg\max_{\Gamma_a^{b,t},\, a \in \mathcal{A}} \langle \Gamma_a^{b,t}, b \rangle. \tag{16}$$

Collecting $\{\Psi_b^t\}_{b \in \mathcal{B}}$ yields our updated set $\Gamma^t$. As in PBVI, we keep at most $|\mathcal{B}|$ vectors in $\Gamma^t$, thus maintaining a point-based approximation.

To better understand the complexity of the backup, let $|M|$ denote the number of categorical atoms used to represent the return distributions, and let $|\hat{R}|$ denote the number of atoms used to represent the reward distributions. DPBVI adds a convolution over reward and return atoms, increasing the per-backup cost from $\mathcal{O}(|\mathcal{B}||\mathcal{A}||\mathcal{O}||V_{t-1}||\mathcal{S}|^2)$ in PBVI to $\mathcal{O}(|\mathcal{B}||\mathcal{A}||\mathcal{O}||\Gamma^{t-1}||\mathcal{S}|^2|M||\hat{R}|)$. This is the only source of additional computation overhead relative to PBVI.

# 6 EXPERIMENTAL RESULTS

Our primary goal in these experiments is to verify that DPBVI learns correct return distributions in the risk-neutral setting. Since PBVI optimizes expected returns, we compare DPBVI and PBVI by evaluating whether the expectation of the learned return distribution $\hat{Z}(b)$ matches the expected value function computed by PBVI at each belief $b$. To further validate distributional correctness, we compare the full return distributions learned by DPBVI against empirical first-visit Monte Carlo (FVMC) return distributions.

## 6.1 RESULTS AND DISCUSSION

Table 1 reports the runtime and iteration counts for PBVI and DPBVI in both variants of the Two-State Noisy-Sensor environment. In all cases, DPBVI converges to the same risk-neutral solution as PBVI but incurs higher computational cost due to distributional backups.

To validate that DPBVI converges to the same value function as PBVI, we examine the maximum relative error between their value functions across belief points over successive backups. Figure 2 shows results for two convergence thresholds: $\epsilon = 1e-3$ and $\epsilon = 1e-6$.

Table 1: Results of DPBVI and PBVI: average runtime and iteration counts for convergence criterion $\epsilon = 1e-3$.

| Environment | Algorithm | Avg. Runtime (s) | # Iterations |
|---|---|---|---|
| Two-State | PBVI | 0.077 | 789 |
| | DPBVI | 9.590 | 789 |
| Stochastic Two-State | PBVI | 0.075 | 769 |
| | DPBVI | 214.451 | 769 |

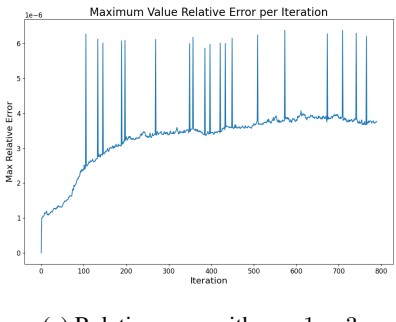
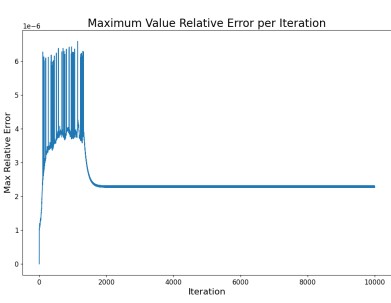

(a) Relative error with $\epsilon = 1e-3$.      (b) Relative error with $\epsilon = 1e-6$.

Figure 2: Maximum relative error between DPBVI and PBVI value functions across iterations in the Two-State Noisy-Sensor environment under two convergence thresholds.

Across both thresholds, we observe very small value differences throughout the run, confirming that DPBVI effectively recovers the same solution as PBVI. Under the standard $\epsilon = 1e-3$ threshold (Figure 2a), the error increases gradually and exhibits intermittent spikes. These spikes are consistent with categorical projection error, floating-point precision limits, and bootstrapping.

To test whether the value function discrepancy is a function of the number of backups—suggesting compounding error or a design flaw in DPBVI—we tightened the convergence threshold to $\epsilon = 1e-6$ and set the iteration limit to $10,000$. As shown in Figure 2b, the error initially increases, peaks below $3 \times 10^{-4}$, and then steadily decreases, plateauing near $5 \times 10^{-5}$. This supports the hypothesis that the observed error stems from numerical and projection effects rather than compounding error over time. Notably, neither PBVI nor DPBVI fully converges under this stricter criterion, yet the discrepancy remains small and bounded.

To validate that DPBVI learns the correct return distributions, we compare its categorical estimates against empirical FVMC return distributions (Section D). Figure 3 shows the comparison for the belief assigning probability $0.368$ to state $s_0$. The mode of both distributions align closely, indicating that DPBVI accurately recovers the underlying return distribution. The longer left tail observed in the FVMC distribution arises naturally from the FVMC procedure: if the belief is first encountered late in a trajectory, even optimal behavior can yield a smaller remaining return. In the true infinite-horizon setting the minimum attainable return is zero (though extremely unlikely), but the finite-horizon FVMC approximation occasionally produces low-return samples, explaining this tail.

The normalized Wasserstein-1 distance between the two distributions is approximately $0.036$, which is the largest discrepancy observed across all beliefs. A full comparison of return distributions for every belief point—including Wasserstein-1 distances—is provided in Section D. Additional implementation details, environment descriptions, and full FVMC results are provided in Appendix B–D.

## 7 CONCLUSION

In this paper, we presented the formal extension of DistRL in partially observable domains. We proved that the partially observable distributional Bellman operator $\widetilde{\mathcal{T}}_{PO}^{\pi}$ and its optimality counterpart $\widetilde{\mathcal{T}}_{PO}^{*}$ are $\gamma$-contractions in the supremum $p$-Wasserstein metric, extending classical DistRL convergence guarantees to POMDPs.

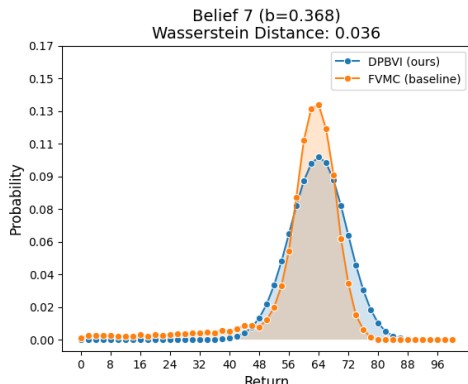

Figure 3: Comparison between DPBVI's learned categorical return distribution (blue) and empirical FVMC distributions (orange) for the belief with probability of $0.368$ of being in state $s_0$. The normalized Wasserstein-1 distance is reported. The learned and empirical distributions closely align, indicating accurate recovery of the return distribution.

We introduced $\psi$-vectors as the distributional analogs of $\alpha$-vectors, proving that under risk-neutral control, the optimal distributional value function in a POMDP admits a finite PWLC representation in the Wasserstein space. This establishes a direct distributional generalization of classical POMDP theory.

Building on these foundations, we described DPBVI, a distributional variant of PBVI. DPBVI applies the distributional Bellman optimality operator at a fixed set of belief points and recovers the same risk-neutral solution as PBVI while maintaining full return distributions. Our experiments confirm this alignment and demonstrate that distributional planning does not compromise solution quality in the risk-neutral regime.

Although our results focus on the risk-neutral setting with known models, modeling the full return distribution under partial observability is a necessary precursor to scalable risk-sensitive control. Prior work has shown that common risk-sensitive operators (e.g., CVaR) fail to be contractions in general (Hau et al., 2023), implying that new operators and finite-representation results are needed—an important direction for future work.

Two extensions follow naturally. First, incorporating unknown dynamics (Uehara & Sun, 2022; Ross et al., 2007) would align theory with the practical use of world model approaches, where latent states approximate beliefs (Zhang et al., 2023; Hafner et al., 2025). Second, integrating risk-sensitive objectives within the distributional POMDP framework would enable safer planning in safety-critical domains where environment models are typically unknown and online interaction is infeasible.

## LLM USAGE

This work is the authors' own. All research questions, algorithm, code, experiments, and proofs were developed and executed by the authors. ChatGPT was used as a writing assistant and as a tool to review prior proof strategies and to explore related work. All literature review, proof development, and validation of related work were conducted by the authors. ChatGPT did not generate novel research ideas or proofs.

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

# A  PROOFS

## A.1  PROOF OF THEOREM 1

**Theorem 1.** *Let $p \in [1, \infty)$ and $\gamma \in [0, 1)$. Consider a POMDP $\mathcal{P}$ in which each reward distribution $P_R(\cdot \mid s, a)$ belongs to $\mathfrak{P}_p(\mathbb{R})$ for all $(s, a) \in \mathcal{S} \times \mathcal{A}$. Then the partially observable distributional Bellman operator $\widetilde{\mathcal{T}}_{PO}^\pi$ is a $\gamma$-contraction on $\mathfrak{P}_p(\mathbb{R})^\Delta$ in the supremum $p$-Wasserstein distance $\bar{w}_p$. That is,*

$$\bar{w}_p(\widetilde{\mathcal{T}}_{PO}^\pi \eta, \widetilde{\mathcal{T}}_{PO}^\pi \eta') \leq \gamma \bar{w}_p(\eta, \eta'),$$

*$\forall \eta, \eta' \in \mathfrak{P}_p(\mathbb{R})^\Delta$.*

*Proof.* We prove contraction of the partially observable distributional Bellman operator $\widetilde{\mathcal{T}}_{PO}^\pi$ by lifting the POMDP $\mathcal{P}$ to its equivalent fully observable belief MDP $\mathcal{M}$, where states are beliefs $b \in \Delta$, actions are $a \in \mathcal{A}$, the reward distribution is $P_R(\cdot \mid b, a) := \mathbb{E}_{s \sim b}[P_R(\cdot \mid s, a)]$, and transitions are governed by the belief update $\tau(b, a, o')$ with probability

$$P(o' \mid b, a) = \sum_{s,s'} b(s) T(s, a, s') \Omega(o' \mid s', a).$$

Assume the following:

- $\Delta$ is a Borel space (as it is a subset of a finite-dimensional simplex).

- $\mathcal{A}$ is finite

- The reward distribution $P_R(\cdot \mid b, a) \in \mathfrak{P}_p(\mathbb{R}) \quad \forall (b, a) \in \Delta \times \mathcal{A}$

- The belief transition kernel $\tau(\cdot \mid b, a)$ is measurable

- The policy $\pi : \Delta \to \mathcal{A}$ is fixed and measurable

Let $\eta, \eta' \in \mathfrak{P}_p(\mathbb{R})^\Delta$ be two distribution-valued value functions. Define the supremum $p$-Wasserstein metric as:

$$\bar{w}_p(\eta, \eta') := \sup_{b \in \Delta} W_p(\eta(b), \eta'(b)), \quad \forall p \in [1, \infty).$$

Fix $b \in \Delta$ and let $(G_b, G'_b)$ be an optimal coupling of $\eta(b)$ and $\eta'(b)$. Now sample the random transition:

- $A \sim \pi(\cdot \mid b)$

- $R \sim P_R(\cdot \mid b, A)$

- $B' \sim \tau(\cdot \mid b, A)$

independently of the couplings $G(\cdot), G'(\cdot)$. Define the random variables:

$$\tilde{G}(b) := R + \gamma G(B'), \quad \tilde{G}(b') := R + \gamma G'(B'),$$

By definition $\tilde{G}_b$ has distribution $\left(\widetilde{\mathcal{T}}_{PO}^\pi \eta\right)(b)$ and $\tilde{G}'$ has distribution $\left(\widetilde{\mathcal{T}}_{PO}^\pi \eta'\right)(b)$. Then:

$$
\begin{aligned}
W_p^p(\widetilde{\mathcal{T}}_{PO}^\pi \eta(b), \widetilde{\mathcal{T}}_{PO}^\pi \eta'(b)) &\leq \mathbb{E}\left[|\tilde{G} - \tilde{G}'|^p \mid B = b\right] \\
&= \mathbb{E}\left[|R + \gamma G(B') - (R + \gamma G'(B'))|^p \mid B = b\right] \\
&= \gamma^p \, \mathbb{E}\left[|G(B') - G'(B')|^p \mid B = b\right] \\
&= \gamma^p \, \mathbb{E}_{B'|B=b}\left[\mathbb{E}\left[|G(B') - G'(B')|^p \mid B = b, \ B'\right]\right] \\
&\leq \gamma^p \, \sup_{b' \in \Delta} \mathbb{E}\left[|G(b') - G'(b')|^p\right] \\
&= \gamma^p \, \sup_{b' \in \Delta} W_p^p\left(\eta(b'), \eta'(b')\right) \\
&= \gamma^p \, \overline{w}_p^p\left(\eta, \eta'\right)
\end{aligned}
$$

Taking the $p$th root on both sides yields

$$W_p(\widetilde{\mathcal{T}}_{PO}^\pi \eta(b), \widetilde{\mathcal{T}}_{PO}^\pi \eta'(b)) \leq \gamma \, \overline{w}_p\left(\eta, \eta'\right).$$

Taking the supremum over $b \in \Delta$, we conclude:

$$\bar{w}_p\left(\widetilde{\mathcal{T}}_{PO}^\pi \eta, \widetilde{\mathcal{T}}_{PO}^\pi \eta'\right) \leq \gamma \bar{w}_p(\eta, \eta'),$$

as desired. $\qquad\square$

### A.2 PROOF OF THEOREM 2

**Theorem 2.** *Let $p \in [1, \infty)$ and let $\widetilde{\mathcal{T}}_{PO}^*$ be the partially observable distributional Bellman optimality operator for a POMDP $\mathcal{P}$. Assume that for every $(s, a) \in \mathcal{S} \times \mathcal{A}$, the reward distribution $P_R(\cdot \mid s, a)$ belongs to $\mathfrak{P}_p(\mathbb{R})$. Suppose there is a unique optimal policy $\pi^*$. Then, for any initial return-distribution function $\eta_0 \in \mathfrak{P}_p(\mathbb{R})^\Delta$, the sequence of iterates*

$$\eta_{k+1} = \widetilde{\mathcal{T}}_{PO}^* \eta_k$$

*converges in the supremum $p$-Wasserstein distance $\overline{w}_p$ to $\eta^{\pi^*}$, the return distribution associated with the unique optimal policy.*

*Proof.* We lift the POMDP $\mathcal{P}$ to its equivalent belief MDP $\mathcal{M}$, where each belief $b \in \Delta$ serves as a fully observable state. The transition dynamics $\tau(b, a, o')$ of the belief MDP are Markovian, and the reward distribution is $P_R(\cdot \mid b, a) := \mathbb{E}_{s \sim b}[P_R(\cdot \mid s, a)]$.

From this point forward, we use $b \in \Delta$ to denote the state of the belief MDP. We assume the following conditions hold in the belief MDP:

- The belief state space $\Delta$ is compact and convex

- The reward distribution is $P_R(\cdot \mid b, a)$ is continuous in $b$ $\forall a \in \mathcal{A}$

- The reward distribution $P_R(\cdot \mid b, a) \in \mathfrak{P}_p(\mathbb{R})$ $\quad \forall (b, a) \in \Delta \times \mathcal{A}$

- The belief update function $\tau(b, a, o')$ is continuous in $b$ for all $a \in \mathcal{A}$ and $o' \in \mathcal{O}$

- The action space $\mathcal{A}$ is finite

- The policy space $\Pi$ is fixed and measurable

- There exists a unique optimal policy $\pi^*$

Let $\eta^{\pi^*}$ denote the return distribution induced by the optimal policy. The partially observable distributional Bellman optimality operator $\widetilde{\mathcal{T}}_{PO}^*$ can be interpreted as a greedy update rule over return distributions in the belief MDP.

Define $Q_\eta(b, a) := \mathbb{E}_{Z \sim \eta(b,a)}[Z]$ as the expected return under the return distribution $\eta$. The *action gap* at state $b \in \Delta$ is defined as

$$\text{GAP}(Q_\eta, b) := \min\{Q_\eta(b, a^*) - Q_\eta(b, a) : a^*, a \in \mathcal{A}, a^* \neq a, Q_\eta(b, a^*) := \max_{a' \in \mathcal{A}} Q_\eta(b, a')\}$$

The global action gap is then defined as

$$\text{GAP}(Q_\eta) := \inf_{b \in \Delta} \text{GAP}(Q_\eta, b)$$

We now justify that $Q_\eta(b, a)$ is continuous in $b$ for all $a \in \mathcal{A}$. The return distribution $\eta(b, a)$ is defined recursively from the reward distribution and belief transitions. Under our assumptions that the reward distribution $P_R(\cdot \mid b, a)$ and belief update $\tau(b, a, o')$ are continuous in b, it follows that $Q_\eta(b, a) = \mathbb{E}_{Z \sim \eta(b,a)}[Z]$ is continuous in $b$ as well. Since the action space is finite, the maximum and second maximum of $\{Q_\eta(b, a)\}_{a \in \mathcal{A}}$ are continuous in $b$, so $\text{GAP}(Q_\eta, b)$ is also continuous in $b$.

By the Extreme Value Theorem, since $\text{GAP}(Q_\eta, b)$ is continuous on the compact state space $\Delta$, the infimum is attained and equal to the minimum. Since the optimal policy is unique, $\text{GAP}(Q_\eta, b) > 0$ $\forall b \in \Delta$, thus $\text{GAP}(Q_\eta) > 0$.

Fix $\epsilon = \frac{1}{2}\text{GAP}(Q^*)$. The standard Bellman optimality operator is known to be a $\gamma$-contraction under the $L^\infty$ norm. Consequently, $\exists k \in \mathbb{N}$ such that

$$||Q_{\eta_k} - Q^*||_\infty < \epsilon \qquad \forall k \geq K$$

For any fixed $b$, let $a^*$ be the optimal action in that state. Then for any $a \neq a^*$, we have that

$$\begin{aligned}
Q_{\eta_k}(b, a^*) &\geq Q^*(b, a^*) - \epsilon \\
&\geq Q^*(b, a) + \text{GAP}(Q^*) - \epsilon \\
&> Q_{\eta_k}(b, a) + \text{GAP}(Q^*) - 2\epsilon \\
&= Q_{\eta_k}(b, a)
\end{aligned}$$

Thus, any greedy selection rule applied to $\eta_k$ will yield the optimal policy $\pi^*$ $\forall k \geq K$. From this point onward, the Bellman updates correspond to evaluation under a fixed policy $\pi^*$, and we may invoke Theorem 1 with the initial return distributions $\eta_0 = \eta_k$ to conclude that $\eta_k \to \eta^{\pi^*}$. Hence, $\widetilde{\mathcal{T}}_{PO}^*$ is a $\gamma$-contraction and converges to the unique fixed point under the supremum $p$-Wasserstein metric. $\qquad\square$

### A.3 PROOF OF THEOREM 3

**Theorem 3.** *Under the assumptions of Theorem 2, the optimal distributional value function $Z^*(b)$ admits a finite piecewise-linear and convex (PWLC) representation under belief-linear mixture with respect to supremum $p$-Wasserstein $\bar{w}_p$. Specifically, there exists a finite set of $\psi$-vectors $\Gamma$ such that for any belief state $b \in \Delta$,*

$$Z^*(b) = \sum_{s \in \mathcal{S}} b(s)\psi^{*,s},$$

*where $\Psi^*(b) \in \Gamma$ is the maximizing $\psi$-vector at belief $b$ and $\psi^{*,s}$ denotes its sth component.*

*Proof.* By Theorem 2, there exists a unique optimal distributional value function $Z^*$. Let $\Gamma^*$ be the set of $\psi$-vectors that are optimal for some belief. For any $\Psi \in \Gamma^*$, define its associated $\alpha$-vector by

$$\alpha(\Psi) \triangleq (\mathbb{E}[\psi^s])_{s \in \mathcal{S}}.$$

Fix a belief $b$ and let $\Psi^*(b) \in \Gamma^*$ be any maximizer of $\langle \Psi, b \rangle = \sum_s b(s)\mathbb{E}[\psi^s]$, where $\psi^s$ is the $s$th component of $\Psi$. Taking expectations yields

$$\mathbb{E}[Z^*(b)] = \sum_s b(s)\mathbb{E}[\psi^{*,s}] = \alpha(\Psi^*(b)) \cdot b,$$

where $\psi^{*,s}$ is the $s$th component of $\Psi^*(b)$. Since $\Psi^*(b)$ is chosen to maximize expected return,

$$\alpha(\Psi^*(b)) \cdot b = \max_{\Psi \in \Gamma^*} \alpha(\Psi) \cdot b.$$

Let $\zeta^* = \{\alpha(\Psi) : \Psi \in \Gamma^*\}$. Then the optimal value function satisfies

$$V^*(b) = \mathbb{E}[Z^*(b)] \quad = \max_{\alpha \in \zeta^*} \alpha \cdot b.$$

By Sondik's result Sondik (1978), $V^*$ is PWLC and can be represented by a finite set of $\alpha$-vectors. Therefore, $\zeta^*$, and hence, $\Gamma^*$, may be taken to be finite. Since $Z^*(b)$ is the belief-linear mixture of the corresponding $\psi$-vectors, the optimal distributional value function admits a finite PWLC representation with respect to the supremum $p$-Wasserstein metric. $\qquad\square$

## B ENVIRONMENTS

We evaluate DPBVI on two variants of a small POMDP environment adapted from (Russell & Norvig, 2020, Section 17.5). Both environments share the same transition and observation structure but differ in their reward models.

**Two-State Noisy-Sensor (Deterministic Rewards)** This is a simple domain with two states, $\{s_0, s_1\}$, and a noisy sensor. The two actions are *Go* and *Stay*. *Go* changes states with probability 0.9 and *Stay* stays in the same state with probability 0.9. The agent receives reward 1.0 whenever it is in state $s_1$ and 0 otherwise. With probability 0.6 the sensor reports the correct state. We set the discount factor $\gamma = 0.99$ and $\mathcal{B}$ to include twenty evenly spaced beliefs between complete belief of being in either state (i.e., $|\mathcal{B}| = 20$).

**Two-State Noisy-Sensor (Stochastic Rewards)** This environment mirrors the transition and observation structure above but uses stochastic rewards. In state $s_0$, rewards follow a Gaussian distribution $\mathcal{N}(1.0, 0.1)$ (truncated to $[0.0, 1.3]$), while for state $s_1$, rewards follow an exponential distribution with rate $\lambda = 10$) (also truncated to $[0.0, 1.3]$). We discretize each reward distribution using forty categorical atoms over this shared support. PBVI receives only the expected reward for each state, while DPBVI learns the full return distributions via their categorical approximations.

## C EXPERIMENTAL SETUP

We disable belief-point expansion in both environments and fix a finite set $\mathcal{B}$ of belief states as described in Section B. PBVI is implemented as in (Pineau et al., 2003), while DPBVI follows the backup procedure from Section 5. Return distributions in DPBVI use 51 categorical atoms, with each $\hat{\psi}$-vector initialized to the uniform distribution, and we adopt a distribution support of $[0, 100]$.

Upon convergence, we compare the expected values of DPBVI's return distributions against PBVI's scalar value function, using identical discount factors and stopping criteria in both algorithms. To assess distributional accuracy, we compute empirical FVMC return distributions for each belief and report the normalized Wasserstein-1 distance between the FVMC baseline and the learned categorical distribution.

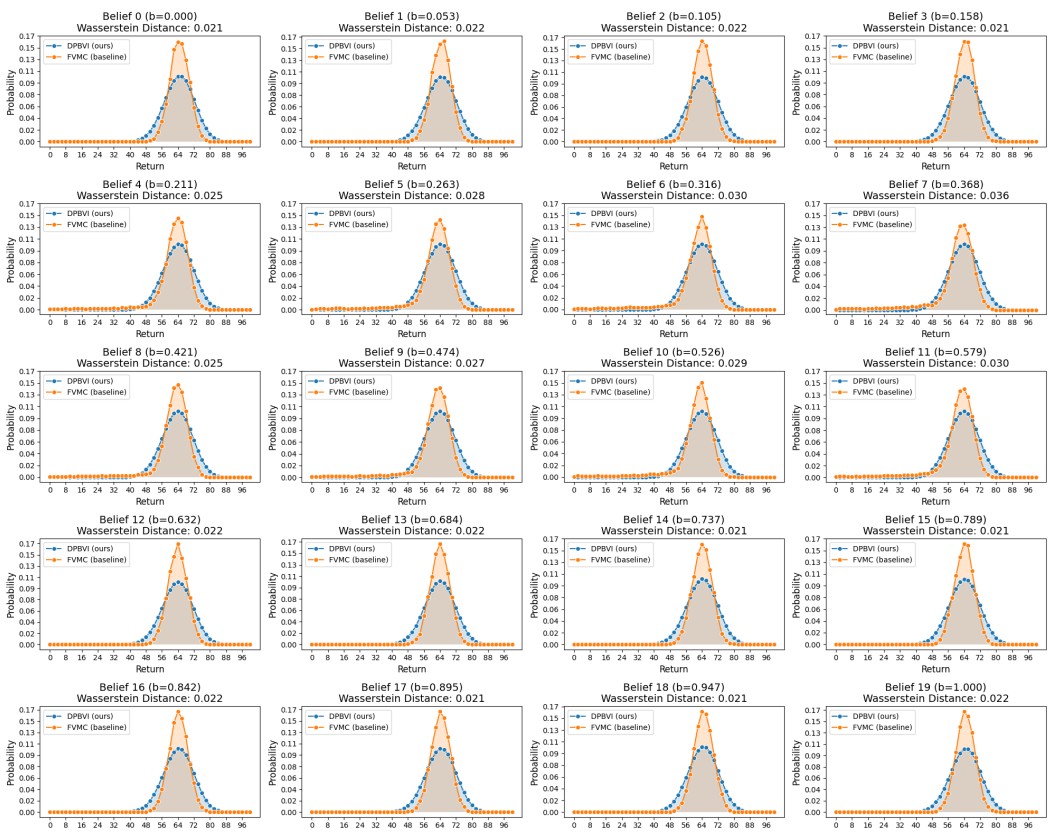

Figure 4: Comparison between DPBVI's learned categorical return distributions (blue) and empirical FVMC distributions (orange) for each belief point in the Two-State Noisy-Sensor: Stochastic Rewards environment. The normalized Wasserstein-1 distance is reported for each belief. Belief values indicate the probability of being in state $s_0$. Across all beliefs, DPBVI closely matches the empirical distributions, validating the correctness of the learned distributions.

## D    FIRST-VISIT MONTE CARLO

To validate the return distributions learned by DPBVI, we compute empirical first-visit Monte Carlo (FVMC) estimates for each belief point. For each belief $b$, we sample $n = 5,000$ trajectories from the true POMDP dynamics and record the discounted return from the first visit to belief $b$ onward. To approximate the infinite-horizon setting, each trajectory is truncated to $T = 1,000$ steps, since $\gamma^{1,000} \approx 0.0$ and future rewards contribute negligibly to the return. The empirical return distribution is obtained from these samples and compared to the corresponding DPBVI categorical return distribution (after projection onto the same support). FVMC is used only as a baseline comparison and does not influence learning.

