# OpenReview forum: "Provable Distributional Value Iteration under Partial Observability"
_ICLR.cc/2026/Conference — Submitted to ICLR 2026_

### Official Review · Reviewer_eBLc · 2025-10-27

**Soundness:** 2
**Presentation:** 1
**Contribution:** 2
**Rating:** 2
**Confidence:** 3

**Summary:**

The authors propose a novel framework that extends distributional RL to the partially observable MDP setting.

**Strengths:**

The problem setting is well-motivated; the combination of partial observability and distributional RL seems like a worthwhile topic to explore. The empirical results are encouraging.

**Weaknesses:**

My biggest concern with this paper is that a combination of incoherent notation and lack of explanation by the authors makes it difficult to fully understand the proposed framework. For instance, consider $Z(b)$, first introduced in line 187, as the return distribution with respect to some belief $b$. Later in the paper, $Z(b)$ seemingly takes on a new interpretation as the value function (i.e. the expectation of the return distribution) with respect to $b$, seemingly contradicting the previous definition (more concretely, consider equation 10, where the resulting $Z(b)$ would be a scalar and not the return distribution itself).

These kinds of notational inconsistencies are scattered throughout the paper, making it difficult to understand the framework. Moreover, even from a conceptual level, the paper lacks clarity. For instance, again consider $Z(b)$ which is never formally defined: are the authors implying that the framework learns a distribution $Z$ with respect to the entire belief distribution $b$ (i.e., a distribution with respect to another distribution)? Or is the intent that the framework learns the distribution $Z$ with respect to $b(s)$ for some state $s$ (i.e., a distribution with respect to a single belief probability)? A formal definition would clarify many of these ambiguities.

Similarly, the flow of the text feels disjointed at times. For instance, consider the claim made in lines 198-201. At first glance, this appears to be a claim made with no justification. Upon further reading, it appears that Theorems 1 and 2 are meant to support this claim, yet this is never explained explicitly to the reader, thereby causing confusion.

Another concern is the empirical analysis. While it is encouraging that DPBVI converges to the same solution as PBVI, it is underwhelming to not have any empirical evidence that shows whether the correct distribution is actually being learnt. For example, in the experiments performed, it is possible that the authors’ method may be learning the wrong distribution that happens to have the same mean as the correct distribution.

Finally, the constant references to risk-sensitive decision-making is unnecessary given that the authors are not proposing any novel risk-sensitive results. In particular, many of the references to the risk-sensitive case ultimately amount to unsupported claims and speculation, which is unhelpful and a distraction to the contributions that the authors are actually making. Distributional RL is a compelling enough field that can stand on its own without needing to involve risk-sensitivity to motivate the need for it. Perhaps even more importantly, the amount of space freed up by removing the risk-sensitive discussion can be used to better explain the authors’ contributions (see my comments above).

**Minor Comments:**
- Line 60: typo: gamma-contraction**s**
- Line 80: citation typo: Astrom shows up twice
- Line 82: typo: finite-
- Line 99: This section is missing early works on distributional RL (i.e., pre Bellemare 2017)
- Some key citations are missing in Section 3. For example, the claim that the belief state serves as sufficient information for the agent to behave optimally in lines 117-118.
- Line 196: $B’$ is never defined.
- Figure 3 in Appendix B is great, and should be in the main text.

**Questions:**

N/A

---

> ### Author Response · Authors · 2025-11-20
>
> Thank you for the detailed feedback and for identifying several areas where clarity can be improved. We address each point below.
>
> 1. Clarifying notation and definition of Z(b)
>
> We apologize for any difficulty caused by our notation. Following Bellemare et al. (2017, 2023), Z(b) is defined as the compound return distribution for a belief b \in \Delta, exactly analogous to how distributional RL uses Z(s) and Z(s,a) in fully observable MDPs. In the POMDP setting, the belief state is the Markovian statistic for planning, so learning Z(b) is the natural extension of classical DistRL notation. We will make this definition more explicit in the manuscript to avoid ambiguity.
>
> 2. Correction to Equation (10)
>
> Thank you for catching this. Equation (10) currently uses the distributional inner product, which returns a scalar (expected value), creating the appearance that Z(b) becomes a scalar. This was incorrect use of notation which resulted in the confusion. Our intent was that Equation (10) selects the ψ-vector associated with the maximizing conditional plan; Z(b) remains the distribution represented by that ψ-vector. We will revise the notation accordingly throughout the paper.
>
> 3. Flow and justification of lines 198–201
>
> These lines were intended to provide intuition for the theory that follows immediately after introducing the operators. We will revise the text to improve the flow of the section.
>
> 4. Empirical illustration of return distributions
>
> We agree that showing the learned return distributions will strengthen the empirical section. In the revision, we will include representative ψ-vector distributions to illustrate that DPBVI learns structured, meaningful distributions and not merely correct expectations.
>
> 5. Discussion of risk-sensitive control
>
> We will scale back the discussion of risk sensitivity. Our intention was to motivate future work, but we agree it should not distract from the core contributions, and we appreciate the suggestion. We will update the manuscript to reflect this.
>
> 6. Minor edits
>
> Thank you for identifying the typos, missing citations, and placement suggestions. We will correct these issues and move the figure from Appendix B into the main text.
>
> Thank you again for the careful reading and helpful suggestions. These revisions will significantly improve clarity and presentation.

---

> > ### Comment · Reviewer_eBLc · 2025-11-25
> > **Response to Author Rebuttals**
> >
> > I thank the authors for their response to my review. For the moment, I maintain my score, though I am open to changing it upon seeing an updated version of the paper.

---

### Official Review · Reviewer_VvxZ · 2025-10-28

**Soundness:** 3
**Presentation:** 3
**Contribution:** 3
**Rating:** 4
**Confidence:** 4

**Summary:**

Summary:

This paper proposes a new value iteration method for partially observable MDPs (POMDPs) under the p-Wasserstein metric, which is to find a policy that maximizes the reward in the worst-case distribution within this Wasserstein ball with p probability. Specifically, the authors consider uncertainty in the reward and transition function, and build on the existence of the convergence guarantees of the value iteration with the Wasstertain metric. The authors show that the value function can be represented as a piecewise linear functions, which also matches the result with standard POMDPs. The authors also demonstrate that their method matches the regular point-based value iteration with no uncertainty or in the risk-neutral case.

My main question is on Theorem 2, where the authors state that the action gap stabilizes after some iteration K, and any greedy action selection would be optimal as the policy will not change after this iteration. Is it possible to determine an order of magnitude of the value K as a function of the number of states, discount factor, and the Wasserstein metric? I think this is the critical proof of the paper, and the authors can further explain and develop this proof in the appendix.

What is the computational complexity of a single DPBVI iteration?  Specifically, what is the order of |M| in this case? Is it a function of the number of beliefs or POMDP states?

The authors state that they do not formally address risk-sensitive objectives and full return distributions, but I think they can still provide some comments or clarifications on what type of risk-sensitive measures both in theory and practice, as this is not mentioned either in the paper or in the numerical results.

Overall, I think this paper is a good theoretical contribution, but it needs clarifications on the proofs and practical examples or reformulations for using POMDPs with practical risk-sensitive measures.

**Strengths:**

The paper makes a solid theoretical contribution to extending standard value functions to risk-sensitive measures using POMDPs and Wasserstein metrics. The proofs and the resulting formulation match the classic POMDP value functions, which is a strength of the paper.

**Weaknesses:**

The paper can be improved by extending the proof of Theorem 2, and also giving comments or numerical examples of solving POMDPs with risk-sensitive measures that can be expressed by the Wasserstein metric.

**Questions:**

My main question is on Theorem 2, where the authors state that the action gap stabilizes after some iteration K, and any greedy action selection would be optimal as the policy will not change after this iteration. Is it possible to determine an order of magnitude of the value K as a function of the number of states, discount factor, and the Wasserstein metric? I think this is the critical proof of the paper, and the authors can further explain and develop this proof in the appendix.

What is the computational complexity of a single DPBVI iteration?  Specifically, what is the order of |M| in this case? Is it a function of the number of beliefs or POMDP states?

The authors state that they do not formally address risk-sensitive objectives and full return distributions, but I think they can still provide some comments or clarifications on what type of risk-sensitive measures both in theory and practice, as this is not mentioned either in the paper or in the numerical results.

---

> ### Author Response · Authors · 2025-11-20
>
> Thank you for the constructive feedback and for highlighting the strengths of the theoretical contributions. We address your questions below.
>
> 1. Order of magnitude for the iteration index K in Theorem 2
>
> Thank you for this suggestion. We will extend the proof sketch in the appendix to include the order of magnitude of K as a function of the discount factor and the action gap.
>
> 2. Computational complexity of a DPBVI backup
>
> We will add a paragraph discussing the computational complexity of a DPBVI backup and its relationship to PBVI, noting that DPBVI is inherently slower due to handling return distributions and is intended as a theoretical reference implementation.
>
> 3. Clarification regarding risk-sensitive measures
>
> We did not include risk-sensitive backups because extending DistRL optimality theory to CVaR and other coherent risk measures in POMDPs is a significant open problem requiring its own operator-level development (recent work, e.g., Hau et al. 2024, shows the augmented-state CVaR approach is not generally valid). Our ongoing follow-up work builds exactly on the theory in this paper to develop risk-sensitive ψ-vector planning. We will add a short discussion of this to avoid confusion. We will clarify that risk sensitivity is outside the scope of this paper.
>
> Thank you again for the helpful suggestions. We will incorporate these improvements in the revision.

---

### Official Review · Reviewer_i97D · 2025-10-30

**Soundness:** 2
**Presentation:** 2
**Contribution:** 2
**Rating:** 2
**Confidence:** 3

**Summary:**

This paper presents a formal extension of distributional dynamic programming to Partially Observable Markov Decision Processes (POMDPs). The authors introduce distributional Bellman operators for partial observability and prove they are contractions under the supremum p-Wasserstein metric. They also propose $\psi$-vectors as a finite, distributional analog to classical $\alpha$-vectors , which represent the return distribution for each state and maintain the piecewise linear and convex (PWLC) property in the Wasserstein space. Building on this, the paper develops Distributional Point-Based Value Iteration (DPBVI), an algorithm that integrates the vectors into the point-based backup procedure. Experimental results show that DPBVI converges to the same risk-neutral solution as classical PBVI, albeit with an order of magnitude increase in computation.

**Strengths:**

- The paper explores an interesting direction: applying distributional dynamic programming to POMDPs. This is a novel perspective that could open up new research avenues.

- The theoretical results, while straightforward, are nice to have for completeness. They confirm that distributional Bellman operators for POMDPs align with those for belief MDPs.

**Weaknesses:**

- The premise is confusing. Distributional RL is fundamentally about learning, but this paper focuses on planning in a known POMDP. This feels more like distributional dynamic programming than RL.

- Theorem 1, 2, and Corollary 1 are not surprising. Since any POMDP can be rewritten as a belief MDP, the extension of distributional RL theory is almost trivial.

- Section 4.2 lacks rigor and clarity. The definition of $\psi$ vectors and $\Psi$ is inconsistent and confusing:

  1. $\psi^s$ is described as a distribution over returns, yet the paper refers to $\psi$-vectors and uses inner products as if these were well-defined vectors.

  2. Equation 9 defines $\Psi$ as a set, not a vector, which contradicts the terminology. Figure 3 suggests discretization via binning. This conflates the general case (the theoretical ground truth) where the return distributions are continuous, with the categorical approximations (like C51).

  3. Convergence guarantees for categorical approximations differ from those for continuous distributions, so Equation 10 and Theorem 3 are questionable. For example, for the categorical approximation, convergence is proven only for Cramer, not p-Wasserstein.

  4. The assumption in Section 5 (“mean-preserving when its support spans all possible returns”) is critical, yet the theory in 4.2 seems to ignore this until the end.

- DPBVI appears to be a straightforward extension of PBVI. Given that Section 4.2 is shaky and other results are trivial, the novelty is limited.

- Experimental results are insufficient and unconvincing: They only show that DPBVI converges to the same result as PBVI in risk-neutral settings, but much slower. No experiments demonstrate any advantage of DPBVI over PBVI, especially in risk-sensitive scenarios. The added computational cost of DPBVI is significant, yet no justification is provided through empirical benefits.

**Questions:**

1. What exactly is the mathematical structure of $\psi^s$ and $\Psi$? Are these vectors, sets, or distributions? The paper needs to make this precise.

2. How can Equation 10 and Theorem 3 hold if $\Psi$ is based on categorical approximations? Isn’t this only approximate, not exact?
What is $\Gamma$ formally? How is the inner product $\langle \Psi, b \rangle$ rigorously defined?

3. Why does the paper use categorical distributions for theoretical development? How does this affect convergence guarantees?

4. Why are there no experiments showing the benefits of DPBVI in risk-sensitive settings? Without this, what is the practical motivation for the approach?

---

> ### Author Response · Authors · 2025-11-20
>
> Thank you for the detailed and thoughtful feedback. We address each concern below and will incorporate clarifications and improvements into the revision.
>
> 1. On “RL vs DP” and the premise of the paper
>
> Distributional RL, like classical RL, is grounded in dynamic programming theory. Foundational results in Distributional RL (e.g., Bellemare et al. 2017) were first developed entirely in the planning / DP setting before being extended to learning algorithms such as C51. Just as with classical RL, developing DistRL theory in the DP setting is a prerequisite to scalable learning algorithms; the DP formulation is not a departure from RL but the theoretical foundation of it.
> Our paper follows this established trajectory: we first extend the distributional Bellman operator and its convergence guarantees to the POMDP setting before moving to scalable and learning-based variants. This is the standard research arc in both RL and DistRL.
>
> 2. On the theoretical results being “unsurprising”
>
> We agree that the results are expected given the belief-MDP equivalence. However, the formal extension of DistRL contraction theory, the PWLC structure, and α-vector/ψ-vector representability has not previously been established for POMDPs.
> This formalization is necessary for the community to build distributional and risk-sensitive planning/learning algorithms in partially observable environments.
>
>
> 3. Clarifying the mathematical structure of ψ and Γ
>
> A ψ-vector is:
> \Psi = (\psi_s)_{s\in S},
> a vector whose components are return distributions (later instantiated numerically via a discretization for DPBVI).
> We will revise Section 4.2 to make this explicit, standardize and unify notation (Ψ as a vector, ψ_s as its components).
> Γ is the analog of the classical set of α-vectors: it contains the current nondominated ψ-vectors produced during PBVI-style backups (see lines 256, 300, and 328). We will explicitly restate this in the revision.
>
>
> 4. On continuous vs categorical distributions
>
> The theory (Theorems 1, 2 and 3) is developed entirely in the space of true return distributions, without assuming any specific finite representation. Theorem 3 follows directly from the operator-level results: it characterizes the PWLC representability of the optimal distributional value function in the abstract distribution space.
>
> The categorical representation used in DPBVI is introduced only in Section 5 as a practical numerical approximation, not a theoretical assumption. We will clarify this more explicitly and adjust Figure 1 to avoid implying ψ-vectors must be categorical.
>
> 5. On mean preservation, projection, and convergence
>
> As stated in Section 5, the categorical representation is mean-preserving when its support spans the return range. Mean preservation is sufficient for PBVI-style DP, because inner products are computed via
> \langle \Psi, b\rangle = \sum_s b(s)\, \mathbb{E}[\psi_s].
> Corollary 1 concerns contraction of the distributional Bellman operator itself. The projection Π referenced in the corollary is a mathematical abstraction used to guarantee the existence of a fixed point. Corollary 1’s projection Π is an abstract operator ensuring existence of a fixed point; it is not instantiated as the categorical projection used in DPBVI.
>
>
> 6. Why we chose the categorical representation
>
> The categorical parameterization is chosen because:
>     1.  it is mean-preserving, which is required for PBVI-style inner products,
>     2.  it can approximate any bounded return distribution with arbitrary precision.
>
> Alternative representations such as quantile approximations are not mean-preserving and therefore are not compatible with ψ-vector PWLC structure. We will clarify this rationale in the manuscript.
>
> 7. On the scope of DPBVI and empirical evaluation
>
> DPBVI is intentionally framed as a reference implementation designed to validate that the proposed theoretical extension operates as expected in practice. Distributional value iteration similarly does not outperform expected-value value iteration; its role is foundational, not empirical. DPBVI’s purpose is to demonstrate correctness, not competitive performance.
>
> While DPBVI follows the structure of PBVI, managing full return distributions (rather than scalars) introduces nontrivial representational and computational considerations. Our goal is to establish the feasibility of ψ-vector backups. Thus, the other aspects of scalability are out of scope for the manuscript.
>
> We will express these concerns by expanding the text to clarify the intent of DPBVI and discuss its computational cost, as suggested.

---

> > ### Author Response · Authors · 2025-11-20
> >
> > 8. On risk-sensitive experiments
> >
> > We did not include risk-sensitive backups because extending DistRL optimality theory to CVaR and other coherent risk measures in POMDPs is a significant open problem requiring its own operator-level development (recent work, e.g., Hau et al. 2024, shows the augmented-state CVaR approach is not generally valid).
> > Our ongoing follow-up work builds exactly on the theory in this paper to develop risk-sensitive ψ-vector planning. We will add a short discussion of this to avoid confusion.
> >
> > Risk-sensitive extensions require establishing contraction of new evaluation and optimality operators under coherent risk measures, which is substantially more complex than the risk-neutral case and cannot be addressed within the space of this paper.
> >
> > 9. Inner product definition clarification
> >
> > We define distributional inner product in section 4.2 line 260 as
> > $\langle \Psi, b \rangle = \sum_s b(s)\mathbb{E}[\psi_s]$
> >
> > Thank you again for the detailed feedback.

---

> > > ### Comment · Reviewer_i97D · 2025-11-25
> > >
> > > Dear authors,
> > >
> > > Thank you for your responses to my review. Unfortunately, I am still not convinced that the paper in its current form has enough contributions to constitute an ICLR paper. For now, I will maintain my score and I am happy to discuss further with my fellow reviewers.

---

### Official Review · Reviewer_hTME · 2025-11-01

**Soundness:** 3
**Presentation:** 4
**Contribution:** 4
**Rating:** 8
**Confidence:** 3

**Summary:**

The authors tackle both state and outcome uncertainty in Partially Observable Markov Decision Processes (POMDPs) by provably extending (for the first time to the best of my knowledge) the Distributional Reinforcement Learning (DistRL) framework and providing convergence guarantees, an important step towards risk-sensitive planning for real-world applications.

This is accomplished by introducing distributional Bellman operators to the POMDP setting and proving convergence to the optimal return distribution (if there is a unique optimal policy) under partial observability (i.e., distributional Bellman operators remain $\gamma$\-contractions under the suprenum p-Wasserstein metric).

The second key contribution is the introduction of $\psi$\-vectors which replace each scalar entry of the standard POMDP $\alpha$\-vector with a distribution over returns. Finite $\psi$\-vectors are then proven to be sufficient to represent the optimal distributional value function while preserving the PWLC property in the Wasserstein space.

The final major contribution is Distributional Point-Based Value Iteration (DPBVI) which adapts PBVI to the distributional setting using $\psi$\-vectors. Under risk-neutral conditions (i.e., reward expectation only), $\psi$\-vectors collapse to $\alpha$\-vectors and key convergence guarantees (Corollary 1) are preserved whilst operating in the space of return distributions. Experimentally, DPBVI is shown to converge ($\epsilon = 1e-3$) to the same value function as PBVI using the MiniGrid DoorKey and Two-state Noisy-Sensor POMDP environments.

**Strengths:**

This paper is well written and provides convincing evidence for each of the core contributions within the defined scope of establishing the theoretical foundations for distributional planning under partial observability. The core significant contributions are detailed in my summary.

I do not know the literature on DistRL well enough to be  certain that this is the first extension of distributional planning to the POMDP setting, but that being the case this appears to be significant and impactful work with good potential for future extensions and real-world applications.

I found the formalisation and proofs to be written in a way that provides a good intuitive understanding alongside the guarantees.

**Weaknesses:**

Whilst the purpose of this paper is to establish theoretical foundations, and appears to do so thoroughly, the experimental results are limited to validating the convergence of DPBVI w.r.t PBVI under risk-neutral / reward expectation only objectives. Given the wider goals of this work to enable risk-aware real-world planning under partial observability, the contribution would be improved by investigating experimentally:

- How DPBVI provides new information, do the $\psi$\-vectors show meaningful return distributions experimentally?

- “We hypothesize these spikes arise from categorical projection error“ - could this be investigated further e.g., by varying the number of bins M?

- DPBVI has a significantly higher runtime “due to the additional cost of distributional operations“. Could this be expanded upon to provide any insight or assurances regarding scalability beyond these small POMDP environments?

**Questions:**

- Do the $\psi$\-vectors show meaningful return distributions experimentally?

- “We hypothesize these spikes arise from categorical projection error“ - could this be investigated further e.g., by varying the number of bins M?

- DPBVI has a significantly higher runtime “due to the additional cost of distributional operations“. Could this be expanded upon to provide any insight or assurances regarding scalability beyond these small POMDP environments?

---

> ### Author Response · Authors · 2025-11-20
>
> Thank you for the thoughtful and constructive feedback. We are glad the theoretical contributions were seen as significant, and we appreciate the suggestions for improving clarity and experimental analysis. We address the points below.
>
> 1. Meaningful ψ-vector return distributions
> We agree that visualizing the learned return distributions would strengthen the contribution. In the revision, we will include figures at representative belief points that compare ψ-vector distributions across the belief simplex. These visualizations will illustrate that DPBVI captures meaningful structure in the return distributions (e.g., multimodality, variance) rather than only matching PBVI in expectation.
>
> 2. Investigating projection-error spikes
> We will examine how the number of return bins M influences the projection-error spikes, and we will incorporate additional results if we observe systematic trends.
>
> 3. Runtime complexity and scalability
> We will add a paragraph discussing the computational complexity of a DPBVI backup and its relationship to PBVI. As the reviewer notes, DPBVI is inherently slower due to handling return distributions; we view it as a theoretical "reference implementation" whose purpose is to empirically validate the formal extension of DistRL to POMDPs.
> We will clarify that practical scalable variants (e.g., deep/approximate ψ-vector models) are a natural direction for future work.

---

### Author Response · Authors · 2025-12-03

We thank the reviewers for their thoughtful and constructive feedback. We addressed the major technical concerns raised and substantially improved clarity throughout the paper.

(1) Reward uncertainty and theoretical updates:
We updated the problem formulation to incorporate uncertainty in rewards and revised theorems, definitions, and proofs in Section 4.1 to ensure correctness under this setting.

(2) Reduced emphasis on risk sensitivity:
We minimized discussion of risk-sensitive extensions to maintain focus on the core risk-neutral distributional results. In the conclusion, we also clarify why we do not tackle risk sensitivity in this work.

(3) Clarified ψ-vectors and finite representation:
We resolved ambiguity in the ψ-vector definition, revised Theorem 3, and added a corollary establishing a finite approximate ψ-vector representation. We also moved the ψ-vector figure to the main text for clarity.

(4) Corrected and clarified DPBVI backup:
Section 5.1 now reflects the revised ψ-vector formalism, and the DPBVI backup equation has been corrected to be consistent with the updated distributional operator.

(5) Strengthened experimental validation:
We expanded the experiments to include empirical first-visit Monte Carlo return sampling, demonstrating that DPBVI learns the correct return distributions and matches PBVI in expectation.

While not every minor suggestion could be incorporated due to space constraints, we believe these revisions address the primary concerns raised by reviewers and significantly improve the clarity, correctness, and completeness of the work.

---

### Meta-Review · Area_Chair_kn2j · 2026-01-04

**Summary:**

The paper proposes a distributional approach to POMDPs, namely, computing the distribution of returns in a partially observable setting.

The reviewers were split regarding the significance of the contribution. Some found that extending the approach to risk-sensitive metrics was necessary to make the contribution sufficient for ICLR, while others considered it an important theoretical step to eventually develop risk-sensitive approaches for POMPDs.

The empirical evaluation also considered only the mean return, which was quite limited and established only equivalence with its standard (non-distributional) counterpart. Such concerns were not addressed during the rebuttal.

**Reviewer Concerns:**

- `hTME` praised the paper, but mentioned that the experiments are very limited and made suggestions on how it could be extended. Given the reply from the authors on December 3, it seems these suggestions were not included in the latest version of the paper.
- `i97D` had multiple concerns with the paper, among them: an unclear reason to link the problem to the distributional RL, numerous inconsistencies that create a lack of clarity, limited novelty, and an unconvincing empirical evaluation. The rebuttal was sufficient to clarify some of the inconsistencies of the paper, but did not address the issues regarding novelty and limited empirical evaluation.
- `VvxZ` noted a lack of experiments with risk-sensitive metrics and suggested some improvements to Theorem 2. The rebuttal did not include new experiments, and it is unclear if Theorem 2 was adjusted accordingly.
- `eBLc` noted a number of inconsistencies in the notation and flow of the paper. Furthermore, `eBLc` found the empirical evaluation insufficient to conclude that the correct distribution of returns was learned besides the mean. The revision includes new results that address the reviewer's concerns about how the estimated distribution of returns approaches the empirical distribution of returns.

**Reviewer Scores:**

- `hTME`: 8 -> 8
- `i97D`: 2 -> 2
- `VvxZ`: 4 -> 4
- `eBLc`: 2 -> 4

---

### Decision · Program_Chairs · 2026-01-26

Reject